# Combined Biological and Numerical Modeling Approach for Better Understanding of the Cancer Viability and Apoptosis

**DOI:** 10.3390/pharmaceutics15061628

**Published:** 2023-05-31

**Authors:** Marko Živanović, Marina Gazdić Janković, Amra Ramović Hamzagić, Katarina Virijević, Nevena Milivojević, Katarina Pecić, Dragana Šeklić, Milena Jovanović, Nikolina Kastratović, Ana Mirić, Tijana Đukić, Ivica Petrović, Vladimir Jurišić, Biljana Ljujić, Nenad Filipović

**Affiliations:** 1Institute for Information Technologies Kragujevac, University of Kragujevac, Jovana Cvijića bb, 34000 Kragujevac, Serbia; msc.katarina.virijevic@gmail.com (K.V.); nevena_milivojevic@live.com (N.M.); katarinapecic13@gmail.com (K.P.); ddjacic@yahoo.com (D.Š.); anamiric53@gmail.com (A.M.); tijana@kg.ac.rs (T.Đ.); 2Faculty of Medical Sciences, University of Kragujevac, Svetozara Markovića 69, 34000 Kragujevac, Serbia; marinagazdic87@gmail.com (M.G.J.); ramovicamra@gmail.com (A.R.H.); n_kastratovic@outlook.com (N.K.); liavaci@gmail.com (I.P.); jurisicvladimir@gmail.com (V.J.); bljujic74@gmail.com (B.L.); 3Faculty of Sciences, University of Kragujevac, Radoja Domanovića 12, 34000 Kragujevac, Serbia; milena.jovanovic@pmf.kg.ac.rs; 4Faculty of Engineering, University of Kragujevac, Sestre Janjić 6, 34000 Kragujevac, Serbia; fica@kg.ac.rs; 5Bioengineering Research and Development Center (BioIRC), Prvoslava Stojanovica 6, 34000 Kragujevac, Serbia

**Keywords:** numerical modelling, cancer, cytostatics, cell viability, apoptosis, gene expression

## Abstract

Nowadays, biomedicine is a multidisciplinary science that requires a very broad approach to the study and analysis of various phenomena essential for a better understanding of human health. This study deals with the use of numerical simulations to better understand the processes of cancer viability and apoptosis in treatment with commercial chemotherapeutics. Starting from many experiments examining cell viability in real-time, determining the type of cell death and genetic factors that control these processes, a lot of numerical results were obtained. These in vitro test results were used to create a numerical model that gives us a new angle of observation of the proposed problem. Model systems of colon and breast cancer cell lines (HCT-116 and MDA-MB-231), as well as a healthy lung fibroblast cell line (MRC-5), were treated with commercial chemotherapeutics in this study. The results indicate a decrease in viability and the appearance of predominantly late apoptosis in the treatment, a strong correlation between parameters. A mathematical model was created and employed for a better understanding of investigated processes. Such an approach is capable of accurately simulating the behavior of cancer cells and reliably predicting the growth of these cells.

## 1. Introduction

Cancer represents a whole group of diseases that include the uncontrolled proliferation of cells with a distinct potential to migrate to other parts of the body [1]. Cancer is one of the leading causes of death worldwide [2]. According to the World Health Organization, cancer was the second leading cause of death worldwide in 2018, accounting for an estimated 9.6 million deaths, or approximately one in six (17%) of all deaths (WHO, Cancer (2020)) [3]. There are many different types of cancer, each with its own set of characteristics and associated treatments. The World Cancer Research Fund estimates that in 2020 there were 1.8 million new cases of colorectal cancer worldwide and 881,000 deaths from the disease (World Cancer Research Fund, Colorectal Cancer Statistics (2021)) [4], while The World Cancer Research Fund estimates that in 2020, there were more than 2.3 million new cases of breast cancer worldwide and 627,000 deaths from the disease (World Cancer Research Fund, Breast Cancer Statistics (2021)) [5]. Early diagnosis and treatment are key to survival [6]. Nowadays, one of the most innovative approaches to predicting cancer fate in patients is using artificial intelligence and a mathematical modeling approach. Artificial intelligence approaches in predicting tumor development and treatment involve the use of algorithms for big data analysis to identify patterns to predict cancer development and progression. These approaches are based on machine learning methods to create predictive models. There are several companies using artificial intelligence in cancer research, such as IBM, Watson Health, Google Health, DeepMind Health, Microsoft Healthcare, and Atomwise. These companies use artificial intelligence to develop new treatments and therapies for cancer, as well as to identify biomarkers for early detection and to help predict the risk of recurrence and development of metastases. In this sense, this publication can be presented as an effort in the direction of therapy prediction.

Mathematical modeling and numerical simulations can be used as an additional tool that could complement the experiments, by providing additional quantitative information about considered cell lines. Numerical models were successfully applied to predict the behavior of tumors both on the cellular [7] and tissue levels [8]. In this study, a numerical model predicting the growth of cancer cells in vitro is applied. The parameters of the numerical model were estimated for all different treatments considered within experiments and afterward, an analysis was performed to observe the effect of considered treatments on the parameters of the model. This analysis provides additional information about the effects of the treatments on the overall behavior of cell lines.

Today, in the treatment of colon and breast cancer, protocols are used that are specifically designed to help in the healing process of patients [9]. Cancer therapy protocols represent a significantly individualized approach to treatment. This may include surgery, radiation therapy, chemotherapy, immunotherapy, and other targeted therapies, often in combination. The goal of cancer therapy protocols is to reduce the risk of recurrence and improve patient outcomes. The most common drugs used in chemotherapy protocols for colon cancer are 5-fluorouracil (5-FU), leucovorin, oxaliplatin, and irinotecan [10]. These drugs are usually administered in combination with other drugs, such as bevacizumab. Other drugs, such as cetuximab and panitumumab, are also used. FOLFOX and FOLFIRI protocols are also often used to treat colon cancer. FOLFOX is a combination of two drugs, 5-fluorouracil and oxaliplatin, while FOLFIRI is a combination of three drugs, 5-fluorouracil, leucovorin and irinotecan. For the purposes of this study, commercial FOLFOX and FOLFIRI cytostatics were used for the treatment of colon cancer. In the treatment of breast cancer, a series of other protocols are used based on the use of other clinical drug protocols: Doxorubicin, Endoxan, Paclitaxel, and Docetaxel [11]. In this study, we also focused on the chemotherapeutics listed above. Biological model systems were human colorectal cancer cell lines (HCT-116), breast cancer (MDA-MB-231) and a healthy cell line isolated from lung pleura (MRC-5). A healthy cell line was treated with 5-FU, leucovorin, oxaliplatin, irinotecan, doxorubicin, endoxan, paclitaxel, and docetaxel.

For the purposes of creating the model, it was necessary to carry out exhaustive biological research. The mechanisms of action of these drugs are generally known, and their mechanisms can be divided into those affecting cell viability [12], processes that induce apoptosis [13] and mechanisms of anti-migratory (anti-metastatic) tumor potential [14]. 5-FU is a drug that works by blocking the production of thymidylate synthetase, an enzyme involved in DNA synthesis, preventing cell reproduction. It is often used in combination with other chemotherapy drugs [15]. Leucovorin is a folic acid analog used in chemotherapy protocols to reduce the toxic effects of anticancer drugs such as 5-fluorouracil. Its action is based on the alteration of folate, which is necessary for DNA synthesis [16]. Irinotecan works by inhibiting the enzyme topoisomerase I, which is involved in DNA replication [17]. Irinotecan is commonly used in combination with other chemotherapy drugs, such as 5-fluorouracil and leucovorin, to improve treatment efficacy. Oxaliplatin is a chemotherapy drug that works by inhibiting the enzyme DNA polymerase, which is involved in DNA replication. Oxaliplatin is also used in combination [18]. Doxorubicin is a type of anthracycline chemotherapy drug used mainly to treat breast and ovarian cancer. It works by incorporating into the DNA of cancer cells and preventing their reproduction [19]. Doxorubicin is commonly used in combination with other chemotherapy drugs, such as paclitaxel and docetaxel, to improve treatment efficacy. Endoxan is a drug that alkylates DNA by interfering with basic replication processes [20]. Paclitaxel is a type of taxane chemotherapy drug used to treat many types of cancer, including breast and ovarian cancer. It works by inhibiting the development of microtubules during cell division [21]. Docetaxel has a very similar mechanism of action to paclitaxel [22]. In addition to the fact that all eight used drugs affect the reduction of tumor viability, especially 5-FU, oxaliplatin and all four-breast cancer chemotherapeutics are well known to induce cell apoptosis. On the other hand, irinotecan and oxaliplatin are also based on anti-metastatic mechanisms. Similarly in the treatment of breast cancer, all four drugs are also involved in anti-metastatic mechanisms. Therefore, it can be concluded that the drug regimens created in this way that are applied in clinical practice cover mostly all three mechanisms in the treatment of cancer.

In this study, we focused on the examination and comparison of the mechanism of cytotoxicity and apoptosis, while the mechanism of influence on the antimetastatic potential was discussed but will be presented in a separate scientific paper. The reason why in this paper we focused exclusively on cytotoxicity and the type of cell death is that the proposed mathematical model was based on cells that are stably attached to the substrate, i.e., on solid tumors that can be classified under stage II, while the mathematical model needed for a better understanding of the process of drug-induced inhibition of cell migration is based on processes closely related to the metastatic potential of tumors, i.e., for cells that are in the phase of migrating from the primary location (stage III and IV). The use of mathematical models for a better understanding of the results of biomedical research opens up a new angle of viewing the problem, which is the subject of this study. Further development of such mathematical models will bring the power of predictability, where on the basis of pre-defined biomedical parameters of an individual patient, it will be possible to successfully make a validated prognosis of the patient’s condition, as well as an optimization of therapy with the aim of the most positive health outcome.

## 2. Materials and Methods

### 2.1. Cell Culturing

In this study, we investigated commercial cytostatics that are kindly donated by the University Clinical Centre Kragujevac, Serbia on two cancer model systems, the HCT-116 cell line of colorectal carcinoma and the MDA-MB-21 cell line of breast adenocarcinoma, and one healthy cell line—MRC-5 fibroblasts isolated from lung pleura. The cells were grown in an incubator with a humidified atmosphere containing 5% CO_2_ at a temperature of 37 °C. All three cell lines were purchased from ECACC. The cells were maintained in Dulbecco’s Modified Eagle Medium (DMEM) (Sigma, D5796) cell culture medium supplemented with 10% fetal bovine serum (Sigma, F4135-500ML) and 1% penicillin/streptomycin (Sigma, P4333-100ML) in culture dishes until satisfying cell confluence.

### 2.2. Chemicals

To perform cell treatments on HCT-116 and MRC-5 cells, 5-FU, leucovorin, irinotecan, and oxaliplatin were utilized, and the stock solutions for these chemotherapeutic agents were 50, 10, 20, and 5 mg/mL, respectively. For MDA-MB-231 and MRC-5 cells, doxorubicin, endoxan, paclitaxel, and docetaxel were employed, and the stock solutions for these chemotherapeutic agents were 2, 40, 60, and 20 mg/mL, respectively.

Treatment concentrations in flow cytometry and qPCR experimentation were used as such—lower concentrations not to be significantly cytotoxic, and higher concentrations to exert cytotoxicity.

### 2.3. Cell Index Viability Assay

Monitoring of the cellular index of the investigated cells was performed using a specific real-time cell analysis on the xCELLigence RTCA DP system (Roche Applied Science, Basel, Switzerland), according to the manufacturer’s instructions. The xCELLigence system consists of three components: (i) RTCA analyzer (determines impedance on gold microelectrodes-E-plate 16); (ii) Threeslot RTCA SP work stage operating unit in which E-boards 16 are placed inside the incubator; and (iii) RTCA Software (version 1.2.1.1002), which can be monitored remotely over an internet connection in real-time. This integrated system for monitoring biological parameters actually serves as a biosensor for monitoring adhesion, i.e., cell viability in real-time. The greater the number of attached (viable) cells on the gold electrode, the greater the measured impedance. The unitless value, cell index (CI), represents quantification of results, describer by the following equation:CI = (R*_tn_* − R_*t*0_)/F
where:

R*_tn_* is the resistance determined at a time t*_n_*,

R_*t*0_ is the resistance determined at the time t_0_,

F is a constant: 15 Ω.

Firstly, we placed 50 µL of DMEM in E-plate 16 wells. After background (blank) estimation, the 10,000 of HCT-116, MDA-MB-231 and MRC-5 cells were added in suspension with a final volume of 100 µL. After cell seeding, the software was programmed to measure the CI index for 96 h in steps (usually every 15 min); 24 h from seeding, the cells were treated with another 50 µL of chemotherapeutics (see concentration range in Results and Appendix A sections). The CI index was monitored until 96 h from seeding or 72 h from treatment. All experiments were done in triplicate. To determine the result of treatment, the software can be used to determine the normalized cell index (NCI) at 24 h, and thus on treatment time. The NCI is an operation function by which the cell index is established as 1.0 (100%). After that, all measuring data are expressed in proportion to this measure. NCI is convenient to approximate the percentage variation in cell adhesion [23].

### 2.4. Flow Cytometry Annexin V/Propidium Iodide (PI) Apoptosis Assay

Flow cytometry apoptosis analyses were performed using 2 × 10^5^ cells per sample. Flow cytometry was conducted on BD FACSVerse Flow Cytometer (BD Biosciences, San Jose, CA, USA). The data were analyzed using the Flowing Software analysis. To quantify the rate of cells in early/late apoptosis and necrosis the cells were cultured in 6-well dishes; 24 h after cell seeding, the cells were treated with chemotherapeutics. After 24 and 72 h from treatment, the apoptosis rate was determined by staining with Annexin V-FITC, followed by the addition of PI (BD Pharmingen, San Diego, CA, USA) according to the manufacturer’s instructions. The subsequent criteria were employed: (1) viable cells were negative to both probes (Annexin V− PI); (2) early apoptotic cells were Annexin V+ PI−; (3) late apoptotic cells were Annexin V+ PI+; and (4) necrotic cells were Annexin V− PI+.

### 2.5. Relative Gene Expression

The 1 × 10^6^ cells were seeded and maintained in 25 cm^2^ culture dishes. After 24 h from seeding, the cells were treated with chemotherapeutics. After 24 and 72 h from treatment, the cells were used for RNA isolation, RNA reverse transcription and qPCR relative gene expression analyses.

#### 2.5.1. RNA Isolation

The samples were homogenized in TRIzol and centrifuged at 12,000 rpm/5 min/4 °C. Chloroform was added to each sample and vigorously mixed for 15 s with subsequent incubation at room temperature for 3 min. Following centrifugation for 15 min/12,000 rpm/4 °C, three phases were separated. The top phase (containing RNA) was transferred to a new microtube, where 500 µL of isopropanol was added and vigorously vortexed. RNA was precipitated for 10 min followed by centrifugation at 12,000 rpm/10 min/4 °C. The supernatant was removed, and the precipitate was washed with 1 mL of 70% ethanol. After ethanol washing the next centrifugation was performed (7500 rpm/5 min/4 °C). The supernatant was discarded, and ethanol was left for drying. Obtained RNA residue was dissolved in 20 µL molecular biology grade water (prepared on Simplicity UV Water Purification System, Millipore, Merck—dual filter 0.45 and 0.05 µm). The RNA concentration was measured on Multiskan Sky UV/Vis spectrophotometer with µDrop plate. RNA indicator purity was estimated as optimal with absorbances (A_260_/A_280_) which were in the range of 1.8–2.0. Samples were further preserved at −20 °C until use.

#### 2.5.2. Reverse Transcription (RT-PCR)

Translation of mRNA into cDNA was performed by using a reverse transcription kit (cDNA Synthesis Kit, Fast Gene Scriptase Basic, Nippon Genetics Japan, Tokyo, Japan); 1.5 μg of total RNA sample in the presence of oligo (dT) primers was added to Master Mix according to the manufacturer’s instructions. The resulting cDNA was obtained at 42 °C for 60 min followed by final incubation at 70 °C for 5 min for enzyme deactivation at Gentier 96E qRT-PCR system (Xi’an Tianlong Science and Technology, Xi′an, China). The cDNA concentration was estimated on the above-described µDrop plate. Samples were further preserved at −20 °C until use.

#### 2.5.3. Quantitative Polymerase Chain Reaction (qPCR)

The produced cDNA was used to evaluate the relative gene expression of the target genes: *Bcl-2*, *Bax*, *Cas3*, *Cas9*, and *Fas*, against the housekeeping gene *β-actin* expression. The commercial kit (FastGene IC Green 2× qPCR Universal Mix, Nippon Genetics Japan) was used for PCR reaction. The reaction mixture was made for each target gene, separately adding forward/reverse primer mix (0.8 μL forward primer and 0.8 μL reverse primer, concentration 10 μM) into 10 μL of qPCR Master Mix, followed by the addition of 1 μg of cDNA and molecular biology grade water as required up to 20 μL. qPCR cycle reaction started with the initial denaturation step, followed by 40 repeated cycles. Each cycle consisted of two steps: (1) DNA denaturation at 95 °C/5 s; (2) annealing and elongation at 60 °C/30 s. Finally, a melting analysis was performed. After amplification, calculation of relative gene expression was calculated by using the Schmittgen–Livak 2^−ΔΔCt^ formula [24]:ΔΔCt = ΔCt1 − ΔCt2.

ΔCt1 = difference between Ct values of the target gene and Ct values of *β-actin* in the drug-treated group; ΔCt2 = difference between Ct values of the target gene and Ct values of *β-actin* in the control group (non-treated cells).

The following genes were evaluated:**Target Name****Abbreviation****Accession Number/Locus****Amplicon Length****Secondary Structures**B-cell CLL/lymphoma 2*Bcl-2*NM_000633.3154 bpNoBcl-2-associated X protein*Bax*NM_138761.4164 bpNoCaspase 3*Cas3*NM_004346.4150 bpNoCaspase 9*Cas9*NM_001229.5243 bpNoFas receptor*Fas*NM_000043.6238 bpNoBeta-actin (housekeeping)*β-actin*NM_001101.51812 bpVery weak

The forward and reverse primer sequences:
Bcl-2F 5′- gataacggaggctgggatgc -3′R 5′- gacttcacttgtggcccagat -3′BaxF 5′- gcttcagggtttcatccagga -3′R 5′- caatcatcctctgcagctcca -3′Cas3F 5′- gcacctggttattattcttggc -3′R 5′- ggactcaaattctgttgccacc -3′Cas9F 5′- actttcccaggttttgtttcc -3′R 5′- caagataaggcagggtgaggg -3′FasF 5′- tggaaataaactgcacccgga -3′R 5′- tcctttctcttcacccaaaca -3′β-actinF 5′- ctcaccctgaagtaccccatc -3′R 5′- aggtctcaaacatgatctggg -3′

### 2.6. Statistical Analysis

For each biomedical analysis, three individual experiments were executed with a minimum of three replicates unless stated otherwise. The data are presented as mean with standard deviation (SD). Statistical analyses were performed using the Mann–Whitney test and one-way analysis of variance (ANOVA).

### 2.7. Numerical Model

In this study, a model developed by Breward et al. [25] was applied. Cancer cells and extracellular material are considered as two independent entities and it is considered that the entire simulation domain contains these two phases that change across the domain over time as cells grow and/or die. The diffusion of oxygen happens across the whole domain and cell death occurs in regions of the domain with low oxygen supply.

If the concentration of cancer cells is denoted by α, the change of concentration over time is governed by the following equation:(1)$$\frac{\partial \alpha }{\partial t}=\frac{\left(1+s_{1}\right)\alpha \left(1-\alpha \right)C}{1+s_{1}C}-\frac{\left(s_{2}+s_{3}C\right)\alpha }{1+s_{4}C}$$
where *C* denotes the oxygen concentration and *s*_1_, *s*_2_, *s*_3_, and *s*_4_ are parameters of the numerical model. The parameter *s*_1_ is related to cell proliferation and hence the first term on the right-hand side of Equation (1) is used to model cell growth. Parameters *s*_2_, *s*_3_ and *s*_4_ are related to cell death, and the second term on the right-hand side of Equation (1) is used to model cell death. More details about the derivation of the equations of the numerical model and the used parameters can be found in the literature [25].

The change in concentration of oxygen across the domain is governed by the following equation:(2)$$\frac{\partial^{2}C}{\partial x^{2}}=\frac{Q\alpha C}{1+Q_{1}C}$$
where the coefficients *Q* and *Q*_1_ are parameters that define the oxygen consumption rate.

Within the original model from the literature [25], the cells were able to move, and additional equations were used for this purpose. Since the focus of this study is on cytotoxicity and apoptosis and during the performed experiments the cells were fixed within the plate, the movement of cells was neglected within the implementation of the numerical model. This enables us to keep the focus on cell death and the parameters related to this phenomenon. The numerical model that also considers the migration of cells will be implemented and used in a separate scientific paper.

The finite element method [26] was used to solve Equations (1) and (2) numerically, by first transforming into incremental–iterative form. The time-dependent simulations in a two-dimensional domain were performed using in-house developed software. The time step was set to 1 h and the total simulation time was set to 72 h, to match the experimental results that included data for an overall 72 h.

An experimental image of control cell lines was used to define the initial cell concentrations at t = 0. The boundary conditions consist in prescribing a zero flux of cell concentration on the boundary of the observed domain (∂α/∂n=0). This was introduced because it was necessary to ensure that in case the cell phase reaches the boundaries, there is an appropriate condition on the solid walls of our chosen domain. It is assumed that the oxygen concentration outside cancer cells is constant (C=const). This is ensured by prescribing the oxygen values in the appropriate nodes of the mesh. Additionally, to ensure that the concentration of oxygen remains constant on the boundaries, the zero gradient boundary condition is defined on the boundaries (∂C/∂x=0). These two conditions were introduced to mimic the experimental conditions, where the cells were grown in aerobic conditions.

The main goal of the numerical simulations within this study is to analyze the influence of diverse treatments on the parameters of the numerical model, in order to subsequently additionally quantify the influence of considered chemotherapeutics on the cancer cells. For this reason, the non-dimensionalization of equations of the numerical model was performed in a way similar to that in the cited literature [25].

The numerical model contains 6 parameters overall: *s*_1_, *s*_2_, *s*_3_, *s*_4_, *Q*, and *Q*_1_. These parameters vary depending on the considered cancer cell line and are estimated independently, by using the data collected during experiments. For the RTCA measurements, overall, 5 time points were used—24, 36, 48, 60, and 72 h since the beginning of the experiment. For the flow cytometry apoptosis measurements, 2 time points were used—24 and 72 h since the beginning of the experiment. The estimation of parameters was carried out in Matlab. The minimization function that was used during the estimation of parameters is given by:(3)$$SE=\sum_{i}{\left(V_{i}^{e}-V_{i}^{s}\right)}^{2}$$
where the considered time points are denoted with the lower index *I*, the experimentally measured percentage of viable cells is denoted by the upper index *e* and the percentage of viable cells obtained in numerical simulations is denoted by the upper index *s*.

The sequential quadratic programming (SQP) method [27,28] was used to determine the area of the global minimum and then the Nelder–Mead simplex optimization algorithm [29,30] was used to determine the exact value of the global minimum.

The values of the parameters of the numerical model vary depending on the applied treatment. The sensitivity analysis was performed to analyze the influence of each parameter on the percentage of viable cells at the end of the observation period (72 h). The results for one cell line (MDA-MB-231 cell line) are shown in Figure 1. For the other two considered cell lines, similar trends were observed. First, the estimation procedure was performed to obtain the values of all 6 parameters for the control MDA-MB-231 cell line. Then, each parameter is varied in a range of ±75%, while the other parameters were kept constant. The graphs in Figure 1 show the sensitivity of the viability to the change of each parameter. The scales on all graphs are set to be equal to better to highlight the influence of each parameter. The percentage of viable cells after 72 h varied in a span of 13.87% for parameter *s*_1_, 151.5% for *s*_2_, 122.18% for *s*_3_, 197.67% for *s*_4_, 4.45% for *Q*, and 5.1% for *Q*_1_. From this analysis, it is evident that the parameters related to cell death have the highest influence on the behavior of cells.

Within the experiments performed in this study, it can be considered that the assumption of aerobic conditions is valid, and the cells can consume as much oxygen as necessary. Within the numerical simulations, it was accordingly considered that an unlimited supply of oxygen is available. For this reason, the parameters related to oxygen consumption have almost no influence on the percentage of viable cells as the sensitivity analysis showed. The conclusion was that the influence of diverse treatments on parameters *Q* and *Q*_1_ could be ignored, and these parameters could be considered fixed for each cell line. Hence, these two parameters were included in the estimation procedure only for the control cell lines and are taken as constant for all other considered cases with treatment.

## 3. Results

### 3.1. Cell Viability Study

In this study, we examined the influence of commercial chemotherapeutics on the viability of colon and breast cancer as well as on healthy fibroblast cells. Depending on the used cell model system, we applied precisely determined chemotherapeutics that are used in clinical practice for the treatment of cancer. In our earlier research, in a similar study, we relied on the processing of cell viability results obtained by a standard biochemical method, where we monitored the reduction of yellow MTT to purple formazan crystals for tracking the metabolic activity of viable cells to estimate the number of cells that survived after drug treatment [31]. In this study, we used a far more sophisticated RTCA method to monitor cell viability at a much larger number of time points in (real) time. In the treatment of HCT-116 cells with 5-FU, we obtained a significant response from the treated cells in terms of a decrease in cell viability in a dose-dependent manner. A similar result of decreased cell viability was observed in HCT-116 cells and the treatment with leucovorin, irinotecan and oxaliplatin (Appendix A). The most significant decrease in viability was obtained in the treatment with 5-FU and oxaliplatin, while leucovorin and irinotecan showed the least significant effect on cell viability. For the sake of comparison with the results of these 4 chemotherapeutics on healthy MRC-5 cells, we observed a similar cytotoxic effect with 5-FU at a lower concentration (10 µM), while the application of a higher concentration (100 µM) (Appendix A) showed significantly lower cytotoxicity in healthy cells compared to colon cancer (Appendix A), which indicates significant selectivity of 5-FU, which is one of the characteristics that make this drug used in clinical practice. Leucovorin shows less selectivity against HCT-116 cells, especially at higher concentrations. Irinotecan shows significantly lower cytotoxicity towards MRC-5 cells and certainly strong selectivity. Perhaps the most significant result was obtained with the use of oxaliplatin, where an exceptional result of selectivity towards HCT-116 cells was obtained. The results of the effects of these drugs on MRC-5 cells are presented in Appendix A, while the effects on colon cancer are presented in Appendix A.

Considering the use of doxorubicin, endoxan, paclitaxel, and docetaxel drugs against MDA-MB-231 breast cancer cells and comparing them with the effect on healthy MRC-5 cells, we also reached similar interesting conclusions. Even the lowest applied concentrations of doxorubicin (0.1 and 1 µM) show a significant cytotoxic effect, while higher concentrations (10 and 50 µM) show an exceptional cytotoxic effect, which is expected in accordance with the nature of the applied drug. Endoxan, on the other hand, shows significantly lower cytotoxic effects, especially in lower concentrations (1 and 10 µM). Paclitaxel and docetaxel show a significant cytotoxic character in a time- and dose-dependent manner on MDA-MB-231 cells (Appendix A). The use of these drugs shows that healthy cells are particularly sensitive to treatment with doxorubicin and docetaxel (especially after 72 h of treatment) (Appendix A). Like MDA-MB-231 cells, endoxan does not show significant cytotoxicity on MRC-5 cells. In the treatment with paclitaxel, MRC-5 cells show significant resistance except in the highest dose after 72 h of treatment.

### 3.2. Apoptosis Rate Analyses—Flow Cytometry

To create a mathematical model for a better understanding and possible prediction of the effect of chemotherapeutics on cancer and healthy cells, it was necessary to additionally analyze the type of cell death. In this sense, our research focused on the methods of proteomics (flow cytometry) and genomics (qPCR). Flow cytometry annexin V/propidium iodide apoptosis assay was used to determine the distribution of subpopulations of cells that are in the phase of apoptosis (early and late), necrosis and viable cells. Appendix A show the results of tests on two cancer cell lines and one healthy cell line. In HCT-116 cells, all 4 applied drugs show a significant effect, especially in the form of induction of late apoptosis in the treatments. Thus, 5-FU (both 10 µM and 100 µM) dominantly affects the occurrence of late apoptosis of cells with a small proportion of necrotic cells and an even smaller proportion of cells in early apoptosis. The share of cells in apoptosis and necrosis shows an increase with the increase in the applied dose and the applied duration of treatment. In comparison, in MRC-5 cells, 5-FU also induces the growth of cells in late apoptosis, but to a very significantly lower extent. The use of leucovorin shows a similar effect on increasing late apoptosis in HCT-116 cells with a large increase in the subpopulation of cells in necrosis. In MRC-5 cells, there is a similar trend, but to a significantly lesser extent. Irinotecan influences the dominant increase in the number of cells in late apoptosis, which increases with increasing dose and applied treatment time, while on MRC-5 this effect is much lower but in a similar trend. Finally, oxaliplatin induces an increase in the proportion of late apoptosis with a similar effect at 10 and 100 µM, but with a significant increase after 72 h of treatment. In MRC-5 cells, oxaliplatin shows a significant increase in late apoptosis to a similar extent as in HCT-116 cells after 24 h, while after 72 h this effect weakens, which indicates an acute effect after which healthy cells recover. In MDA-MB-231 cells, all four selected drugs show a similar effect: an increase in the proportion of late apoptosis and necrosis with an increase in the dose and time of exposure of the cells to the treatment. In the applied testing conditions, the most significant effect is shown by paclitaxel and docetaxel with a large proportion of necrosis in the subpopulation of cells. The comparison of the effect in MRC-5 cells for these 4 drugs indicates a similar trend as in MDA-MB-231 cells only to a significantly lower extent and with a significantly lower proportion of necrosis.

### 3.3. Apoptosis Pathway Investigation—qPCR Approach

For a better understanding of the molecular mechanisms of the detected apoptosis process in the treatment with the proposed chemotherapeutics, the gene parameters involved in these processes were examined. In this sense, a set of genes that are standardly involved in the processes of apoptosis were analyzed: *Bcl-2*, *Bax*, *Caspase3*, *Caspase9*, and *Fas*. In HCT-116 cells (Appendix A) *Bcl* and *Bax* genes show increased expression indicating that these drugs induce apoptosis in a dose- and time-dependent manner. If we consider the absolute numbers and compliance with other methods, we conclude that this increase lasts for the vast majority of treatments, but not for all; however, it is important to notice that the biological evaluation trend is in more significant accordance. This increase is not uniform Irinotecan and oxaliplatin, in particular, extremely strongly induces an increase in the expression of these two genes. In MRC-5 cells, on the other hand, we also observe a strong gene response in terms of increased expression of these two genes, especially with irinotecan and 5-FU. In MDA-MB-231c cells, the expression of *Bcl-2* and *Bax* genes is generally more strongly expressed compared to HCT-116 cells (except in the treatment with docetaxel). Drugs for the treatment of breast cancer applied to healthy cells also indicate a significant induction of transcription of these two genes but to a lesser extent than in cancer cells. By analyzing the second group of genes (*Cas3* and *Cas9*), we focused on elucidating the question of the path of initiation of the apoptosis process. Thus, we concluded that *Cas3* in HCT-116 shows a moderately increased expression in a dose and time dependent manner, while *Cas9* mainly shows a strong but acute effect, i.e., higher expression after 24 h (Appendix A). In comparison, in MRC-5 cells these 4 drugs show even more significant expression of *Cas3*, while *Cas9* shows fluctuating and lower expression. Similarly, in MDA-MB-231 cells, the investigated drugs induce the expression of these two genes generally in a dose- and time-dependent manner. On the other hand, drugs for the treatment of breast cancer used in healthy MRC-5 cells also show a strong effect on the expression of these 2 genes. Finally, after examining genes responsible for the execution of apoptosis (*Bcl-2*, *Bax*) and genes indicating a strong intrinsic apoptosis pathway (*Cas3*, *Cas9*), we also examined the gene expression of the *Fas* receptor responsible for the extrinsic apoptosis pathway. The results on all three cell lines indicate a dominant reduction or slight fluctuation of *Fas* gene expression, which indicates that the extrinsic apoptosis pathway is not favored in these treatments.

### 3.4. Numerical Simulations

As it was mentioned in the Materials and Methods section, the parameters Q and Q1 are estimated only for the control cell lines. For the HCT-116 cell line, the nondimensional values of parameters Q and Q1 are equal to 5.32 and 4.97, respectively. For the MDA-MB-231 cell line, the values of the mentioned parameters are 5.31 and 4.29, while for the MRC-5 the values are 4.69 and 3.98.

The values of estimation error for all considered cases for all cell lines vary from 0.03 to 0.5. The estimation procedure was first performed using the RTCA experimental values. The estimated values of parameters s_1_–s_4_ from Equations (1) and (2) are shown in Figure 2, for the MDA-MB-231 cancer cell line for all considered drug treatments. Then, the estimation procedure was repeated only for the extracted RTCA values and simultaneously for the flow experimental values. The results for the estimation of parameters are shown in Figure 3, again for the MDA-MB-231 cancer cell line for all considered drug treatments. The same results for the other two considered cell lines can be found in the Appendix A. Since the main goal of the numerical simulations in this study was to quantitatively analyze the influence of considered treatments on the change of parameters of the numerical model and accordingly on the behavior of cells, the graphs in Figure 2 and Figure 3 are plotted with relative values of the parameters. The percentage of change of parameters refers to the relative percentage of the values of parameters for each considered treatment with respect to the parameters obtained for the control cell line. As can be concluded from the sensitivity analysis, smaller values of parameters s2 and s3 and higher values of parameter s4 indicate that the cytotoxicity of the applied chemotherapeutic is higher. For the most considered chemotherapeutics, this trend can be observed. The higher concentrations of drugs cause a greater change in the values of the parameters of the numerical model. In that sense, the results of numerical simulations show a trend similar to the one noted during the analysis of experimental findings and are in accordance with the conclusions drawn from the experimental findings.

The change of NCI over time for the control MDA-MB-231 cell line and one considered treatment with paclitaxel 100 µM is shown in Figure 4. The lines on the graph show the experimentally measured values of NCI, using the RTCA technique, as it was already explained in Section 2.3. The dots on the graph show the results obtained in numerical simulation. As can be observed, the results of numerical simulations agree well with experimental results. The value of the SD error for the control cell line is equal to 1.77% and the SD error for the considered treated cancer cell line is equal to 3.48%. An overall observation can be made that the number of cells for the control cell line increases much faster and in time there are many more cells than at the beginning of the observation period. On the other hand, after the treatment with Paclitaxel, the change in the number of cells has a totally different trend, and there is a significantly smaller increase in the number of cells.

## 4. Discussion and Conclusions

In our biological research, which is strongly based on the logic of medical practice, we obtained results that correlate to a significant extent. Cell viability was investigated using the RTCA method, while the type of cell death was investigated using flow cytometry. Apoptosis molecular pathways were estimated by relative gene expression. These three methods were chosen for two reasons: (i) compatibility of cytotoxicity and cell death type assays and (ii) precise mathematical modeling requires a lot of data. RTCA is a sophisticated method for monitoring cell viability in real-time. However, since the RTCA signal correlates with the number of viable cells, this method only detects cells that are stuck to the gold electrodes and we cannot estimate the state of those cells, except to assume that all stuck cells are still alive. This makes sense; however, the literature suggests that cells in early apoptosis still retain active surface adhesion mechanisms [32]. Apoptosis is a form of programmed cell death that is regulated and involves a series of changes in the cell’s outer membrane and cytoskeleton. During this process, cells become more rounded and may attach to surfaces due to the weakened cytoskeleton and increased surface area. Some studies indicate that even cells in late apoptosis still can adhere to surfaces. The fate of a cell that is in early/late apoptosis at the moment of examination changes significantly at the next moment, i.e., the cell dies. qPCR was used to investigate the molecular mechanisms of induction of the examined cancer cells into apoptosis. Treatment concentrations in flow cytometry and qPCR experimentation were used as such—lower concentrations not to be significantly cytotoxic, and higher concentrations to exert cytotoxicity. Our previous results [31] on the cytotoxicity of these chemotherapeutics enabled us to choose these two concentrations. For example, doxorubicin treatment concentrations were significantly lower because of extreme toxicity on MDA-MB-231 cells—IC_50_^24h^ = 37.9 µM, IC_50_^72h^ = 0.2 µM [31]. RTCA treatment concentrations for doxorubicin were 0.1, 1, 10, and 50 µM, while for flow cytometry and qPCR they were 0.1 and 1 µM.

The results obtained by the RTCA method significantly coincide with the results obtained by the MTT method [31] with an even stronger correlation in relation to the trend of the influence of selected chemotherapeutics. Despite the significant selectivity of the applied drugs in relation to cancer/healthy cells described above, which was expected, some toxicity was still observed on healthy MRC-5 cells. The highest toxicity towards healthy cells was shown by 5-FU and oxaliplatin, which fit into their mechanisms of action. Irinotecan and leucovorin generally have the least isolated effect on cells because they are used as adjunctive drugs to increase the efficacy and at the same time reduce the side effects of 5-FU and oxaliplatin in combined protocols. When comparing the toxic effect of selected chemotherapeutics on breast cancer and healthy cells, we reach a similar conclusion as with HCT-116 cells. Except for doxorubicin, which is generally a very toxic substance, the remaining three drugs show significant selectivity towards MDA-MB-231 cells, of course, except for extreme treatment conditions (very high doses and length of treatment time). This is also understandable and in clinical practice is defined within the very broad term of side effects or adverse drug reactions.

It is not surprising that 5-FU, irinotecan and Ox-Pt exhibit significant selectivity towards cancer cells; 5-FU interferes with DNA synthesis and repair, leading to the inhibition of cell proliferation [33]. It is preferentially taken up by rapidly dividing cells, including cancer cells, and is incorporated into the DNA, where it inhibits the activity of thymidylate synthase, an enzyme essential for DNA synthesis. Cancer cells are more dependent on the rapid replication of DNA and RNA than normal cells, making them more susceptible to the effects of 5-FU. Irinotecan is a topoisomerase inhibitor that prevents DNA replication and transcription by inhibiting the activity of topoisomerase I, an enzyme essential for DNA unwinding during these processes [34]. Cancer cells have higher levels of topoisomerase I than normal cells, making them more susceptible to the effects of irinotecan. Additionally, irinotecan is converted in the liver to an active metabolite, SN-38, which is highly toxic to cancer cells [35]. Oxaliplatin is a platinum-based chemotherapy drug that forms adducts with DNA, leading to the inhibition of DNA replication and transcription. Cancer cells are more susceptible to the effects of oxaliplatin due to their higher rate of DNA replication and transcription compared to normal cells [36]. Additionally, oxaliplatin can induce apoptosis, a programmed cell death mechanism, in cancer cells. Like colorectal cancer treatment agents, breast cancer therapy drugs paclitaxel, docetaxel and endoxan also exerted some cancer selectivity. Paclitaxel and docetaxel belong to a class of drugs known as taxanes, which interfere with microtubule dynamics and inhibit cell division [37]. They bind to the beta-tubulin subunit of microtubules and stabilize them, leading to the inhibition of cell division and ultimately, cancer cell death. Cancer cells are more dependent on microtubules for cell division and maintenance of their shape compared to normal cells, making them more susceptible to the effects of taxanes. Cyclophosphamide is an alkylating agent that damages DNA by cross-linking the DNA strands, leading to the inhibition of DNA replication and transcription. Cancer cells are more susceptible to the effects of cyclophosphamide due to their higher rate of DNA replication and transcription compared to normal cells [38]. Additionally, cyclophosphamide can induce apoptosis in cancer cells. Overall, these drugs exhibit selectivity towards cancer cells despite healthy cells due to their different mechanisms of action, which target processes that are more essential for the survival and proliferation of cancer cells. However, these drugs can also affect normal cells, leading to various side effects, and their use requires careful monitoring and management by healthcare professionals.

By analyzing the proportion of early/late apoptosis and necrosis using the flow cytometry method, we came to results that point to (i) an increased effect on the increase in late apoptosis and necrosis (especially in MDA-MB-231 cells) and (ii) significant selectivity against cancer cells. If we consider that cells in early apoptosis still attach smoothly to the substrate and if we know that cells in early apoptosis still show enzymatic activity, we can consider them viable for the purposes of this study. Given that the RTCA method detects only attached cells and defines them as alive, for the purposes of comparing the effect of the investigated drugs and the correlation between RTCA and Flow cytometry analyses, we consider that cells in early apoptosis are attached and viable too. When these relationships are considered in Appendix A and compared, on the one hand, the viable and early apoptosis group with the late apoptosis and necrosis group, two groups of cells are obtained. The first has almost unhindered enzymatic activity and can stick to RTCA microelectrodes, while the second group cannot stick and give a positive signal to RTCA. By such a comparison, we obtain a significant correlation in trend (not in absolute values) between the real-time viability determination method and the cutoff results of determining the type of cell death. Although the proposed mathematical model independently analyzed each individual subpopulation of cells and compared them with RTCA values, we believe that this angle of viewing the results is useful for a better understanding of the impact of chemotherapeutics. Considering that the influence in vivo is very dynamic, we also record similar dynamics and make sections in certain time periods.

The qPCR results of relative gene expression did not enter the creation of the proposed mathematical model but only serve for a better understanding of cell apoptosis pathways. The rate of relative gene expression agrees with registered apoptosis, but here it is important to follow the trend of increase or decrease in gene expression, rather than the ratio of absolute numbers. The reason for this interpretation is the existence of a whole set of additional regulatory cellular mechanisms that occur after the transcription process: post-transcriptional processing, translation, and posttranslational modifications. These post-transcriptional mechanisms are complex, dependent on many factors, and require additional comprehensive cellular proteomics studies for full understanding, which is not the goal of this study. The obtained results indicate a strong expression of *Bcl-2* and *Bax*, responsible for the apoptosis process triggering, and the *Cas3* and *Cas9* genes, which indicate that apoptosis is dominantly defined by the intrinsic initiated pathway. On the other hand, *Fas* indicated extrinsic apoptosis initiation was significantly reduced or without significant changes.

The molecular mechanisms of induction of apoptosis could be explained by chemotherapeutics. The apoptotic mechanism of action for each of these drugs is different; 5-FU inhibits the activity of thymidylate synthase, an enzyme essential for DNA synthesis, leading to DNA damage and the activation of the p53 pathway, a well-known tumor suppressor pathway that regulates apoptosis [39]. The activation of p53 leads to the upregulation of pro-apoptotic genes, such as Bax and Puma, and the downregulation of anti-apoptotic genes, such as Bcl-2, ultimately leads to the activation of the caspase cascade and apoptosis. Irinotecan inhibits the activity of topoisomerase I, an enzyme essential for DNA replication and transcription, leading to DNA damage and the activation of the p53 pathway [40]. Oxaliplatin form adducts with DNA, leading to the inhibition of DNA replication and transcription and the activation of the p53 pathway. Additionally, oxaliplatin can activate the mitogen-activated protein kinase (MAPK) pathway, which can also induce apoptosis [37]. Similar to colon cancer drugs, paclitaxel, docetaxel, and cyclophosphamide (Endoxan) are chemotherapeutic agents that induce apoptosis in cancer cells through similar mechanisms of action. The inhibition of cell division can lead to the activation of the p53 pathway [40]. Overall, paclitaxel, docetaxel, and cyclophosphamide induce apoptosis in cancer cells through the activation of the p53 pathway and the caspase cascade.

The mechanism of induction of apoptosis of colorectal cancer therapy by molecular mechanisms via caspase cascade, Bcl-2, and Bax has been shown in several studies [41,42,43]. Similarly, it is known that doxorubicin, paclitaxel, endoxan, and docetaxel therapy also, among other things, induce apoptosis processes in breast cancer treatment through the above-mentioned mechanisms [44,45,46,47].

Throughout this paper, the nondimensional values of the parameters of the numerical model were presented. This way it is possible to analyze generic parameters that are used to quantitatively describe the behavior of diverse cell lines and to discuss the rate of their change with respect to different applied treatments and different cell lines.

The model used within this study provides the results of cell viability that agree well with experimentally measured values. This proves that this model can be used to accurately predict the behavior of cell lines in vitro. The next step in the process of future improvements of the presented model will be to use the values of estimated parameters for considered concentrations of pharmacological treatments as a basis to predict the values of parameters for other concentrations of these treatments. These values of parameters will then be used to predict the change in cell viability over time for these new cases. Of course, these results would then also require appropriate experimental validation, but this improved model would open a new venue for performing in silico studies of various drug concentrations and finding an optimal value of concentration that should be applied.

The main aim of applying numerical simulations within the research of cancer progression and treatment is to develop a model capable of accurately simulating the behavior of the cancer cells and reliably predicting the growth of these cells. Using this model, it would be possible to analyze a wide variety of possible ways of therapeutically targeting different aspects of cancer cells in silico, without the need to perform time-consuming and costly experiments for every considered treatment. The model used within this study is one step towards this goal.

Cancer is a complex disease characterized by uncontrolled cell growth and proliferation, which can result in the formation of tumors and metastasis. The understanding of cancer viability and apoptosis, a programmed cell death mechanism, is crucial for developing effective therapies against cancer. To better understand this relationship, both biological and numerical modeling approaches have been used. The biological approach involves experimental methods such as cell culture, flow cytometry, relative gene expression, and cell viability monitoring in real-time to study the behavior of cancer cells under different conditions. These experiments can help identify key signaling pathways and molecular mechanisms involved in cancer viability and apoptosis. On the other hand, numerical modeling approaches use mathematical and computational models to simulate the behavior of cancer cells and predict their response to different stimuli. These models can consider various factors such as the cellular microenvironment, gene expression patterns, and signaling networks to provide insights into cancer behavior at a systems level. Combining both approaches can provide a comprehensive understanding of cancer viability and apoptosis and aid in the development of personalized cancer therapies. Our mathematical model can help integrate complex and heterogeneous data from different sources, including genomics, proteomics, and transcriptomics. This integration can provide a more comprehensive understanding of the underlying mechanisms of cancer viability and apoptosis. On the other hand, a mathematical model can help identify key signaling pathways and molecular targets that are involved in the regulation of cancer viability and apoptosis. This information can aid in the development of new cancer therapies and drug targets. Additionally, a mathematical model can be used to predict the response of cancer cells to different treatments and identify potential drug combinations that could enhance their effectiveness. Finally, mathematical models can be used to simulate the behavior of cancer cells in different microenvironments and predict their behavior in vivo. This information can aid in the development of personalized cancer therapies and improve the efficacy of current treatments. Overall, the use of mathematical models in understanding the relationship between cancer viability and apoptosis is important because it can provide a more comprehensive and quantitative understanding of this complex biological process, and ultimately aid in the development of new cancer therapies. Based on the results of biomedical research on the impact of eight chemotherapeutics on cell model systems, we come to three significant conclusions: (1) a strong effect on increasing late apoptosis and necrosis in cancer models; (2) significant selectivity towards cancer in relation to healthy cells; and (3) a strong correlation between certain parameters of viability and apoptosis.

## Figures and Tables

**Figure 1 pharmaceutics-15-01628-f001:**
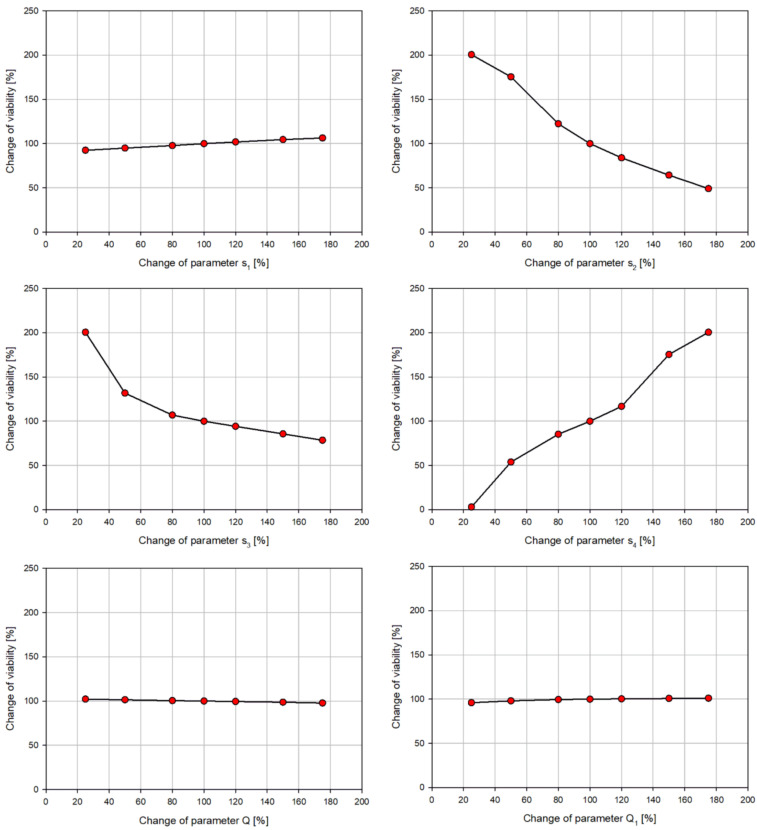
Sensitivity analysis of the influence of change of individual parameter values on the percentage of viable cells for the MDA-MB-231 cell line.

**Figure 2 pharmaceutics-15-01628-f002:**
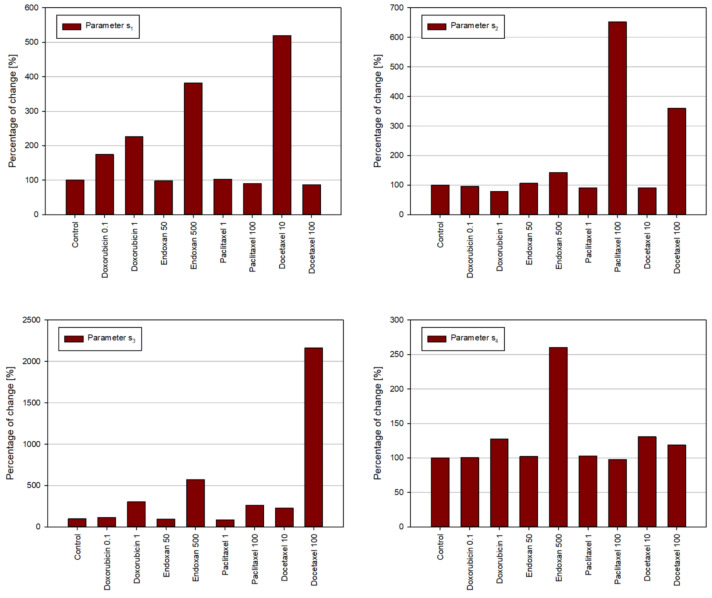
Results of the estimation procedure for the MDA-MB-231 cell line, for all considered treatments, using RTCA experimental data. The percentage of change of parameters refers to the relative percentage of the values of parameters for each considered treatment, with respect to the parameters obtained for the control cell line.

**Figure 3 pharmaceutics-15-01628-f003:**
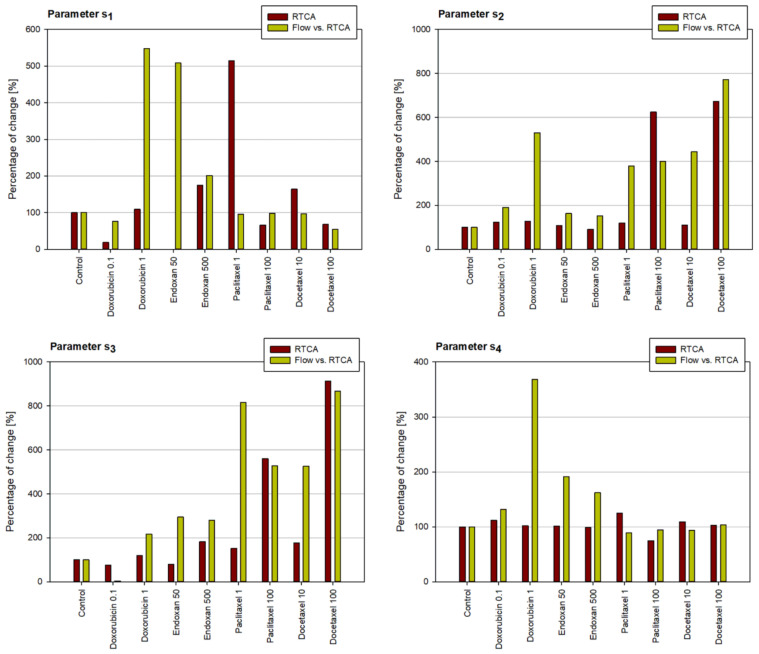
Results of the estimation procedure for the MDA-MB-231 cell line, for all considered treatments, using simultaneously RTCA and flow experimental data. The percentage of change of parameters refers to the relative percentage of the values of parameters for each considered treatment, with respect to the parameters obtained for the control cell line.

**Figure 4 pharmaceutics-15-01628-f004:**
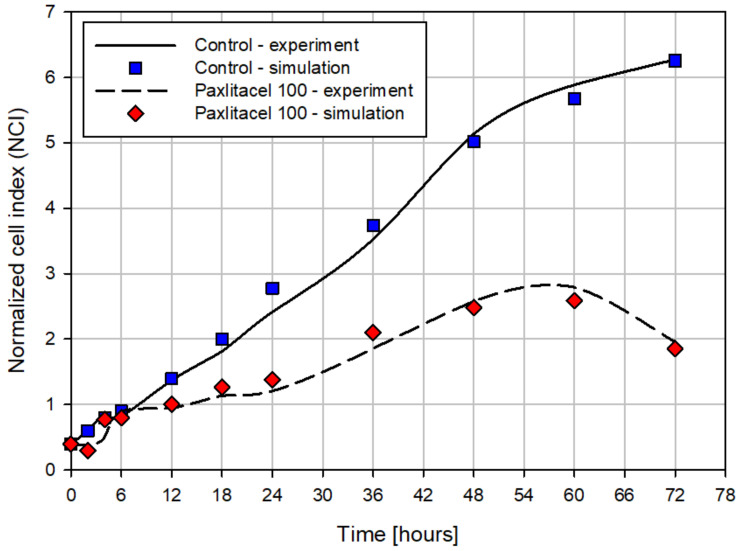
Change of normalized cell index over time for two considered cases for the MDA-MB-231 cell line. The experimental data were measured using the RTCA technique.

## Data Availability

Data is contained within the article.

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
