# Peer review of "Combined Biological and Numerical Modeling Approach for Better Understanding of the Cancer Viability and Apoptosis"

_pharmaceutics, 2023, doi:10.3390/pharmaceutics15061628_

Round 1
Reviewer 1 Report
I find the work presented in the manuscript to be interesting. However, I would like to make some suggestions for improvement. Please consider the following remarks and questions:
1. Was the study submitted to the ethics committee before conducting the tests on cells? Please include this information in the materials and methods section.
2. lines 143-148: It would be helpful if you could reformulate all reactants in a single paragraph and avoid repetition.
3. lines 152-155: the example of doxorubicin should be mentioned in the discussion part.
4. In line 173, it is stated that "F is a constant: 15 Ω at 10 kHz, 12 Ω at 25 kHz, and 10 Ω at 50 kHz." Please clarify that the unit of F and Ohm is not kHz.
5. line 190: Reference [24] is not appropriate for cytometry material. Please delete it as it is already referenced in the previous paragraph.
6. Please review the formula in line 240, as there may be an error.
7. It would be beneficial to include error bars in Figures 1, 2, and 3.
8. Please include standard deviations with the mean in tables 1 and 2.
9. Please correct the typo in line 497 from "tima" to "time".
10. It would be best to separate the discussion and conclusion sections.
11. Please add more information about the Biological and Numerical Modeling Approach for Better Understanding of the Cancer Viability and Apoptosis relationship.
Thank you for considering my suggestions.
Sincerely,
Moderate editing of English language
Author Response
I find the work presented in the manuscript to be interesting. However, I would like to make some suggestions for improvement. Please consider the following remarks and questions:
- Was the study submitted to the ethics committee before conducting the tests on cells? Please include this information in the materials and methods section.
- We fulfilled this information in “Institutional Review Board Statement”.
- lines 143-148: It would be helpful if you could reformulate all reactants in a single paragraph and avoid repetition.
- Reformulated according to suggestions lines 143-147
- lines 152-155 (151-154): the example of doxorubicin should be mentioned in the discussion part.
- Example mentioned in Discussion part.
- In line 173, it is stated that "F is a constant: 15 Ω at 10 kHz, 12 Ω at 25 kHz, and 10 Ω at 50 kHz." Please clarify that the unit of F and Ohm is not kHz.
- finally, we consulted the manufacturer's original protocol and modified the claim, leaving it as F is 15 Ω.
- line 190: Reference [24] is not appropriate for cytometry material. Please delete it as it is already referenced in the previous paragraph.
- Deleted
- Please review the formula in line 240, as there may be an error.
- Reviewed and corrected
- It would be beneficial to include error bars in Figures 1, 2, and 3.
- The parameters were estimated once for each considered treatment and hence there are no replicates. These values of parameters were exact, without calculated errors for individual parameters.
- Please include standard deviations with the mean in tables 1 and 2.
- The parameters were estimated once for each considered treatment and hence there are no replicates. These values of parameters were exact, without calculated errors for individual parameters.
- Please correct the typo in line 497 from "tima" to "time".
- Corrected
- It would be best to separate the discussion and conclusion sections.
- According to the Instructions for authors we decided to merge these two sections. Also, previously accepted publications for this special issue also combined these sections. We hope this is not a crucial obstacle, but if necessary we will change it based on your suggestions.
- Please add more information about the Biological and Numerical Modeling Approach for Better Understanding of the Cancer Viability and Apoptosis relationship.
- Explained in more detail in Discussion and Conclusion part
Reviewer 2 Report
1. Does the selectivity of anti-cancer drugs to healthy or cancer cells depend on the mechanism of action of each drug? Please discuss.
2. Also, does the mode of cell death depend on the mechanism of action on each anti-cancer drug/group? Please discuss.
3. Section 3.3, are there any other supportive data from previous publications to compare with current data?
4. What are parameters s1-s4, specifically?
5. Tables and Figures that contain similar data are not necessary. Please leave only Table of Figure.
6. What is the trend of Figure 4? Please also clarify the definition of normalized cell index.
Overall is ok. May check spelling again.
Author Response
- Does the selectivity of anti-cancer drugs to healthy or cancer cells depend on the mechanism of action of each drug? Please discuss.
- Extended paragraph explaining the above mentioned comments added to Discussion and Conclusion section
- Also, does the mode of cell death depend on the mechanism of action on each anti-cancer drug/group? Please discuss.
- Explained in Discussion and Conclusion section
- Section 3.3, are there any other supportive data from previous publications to compare with current data?
- Discussion regarding this suggestion is added, but in Discussion and Conclusion section.
- What are parameters s1-s4, specifically?
- Parameters s1-s4 are contained within the numerical model. Parameter s1 is used to characterize cell growth, while the other three parameters are used to characterize cell death. These parameters are defined within the numerical model that was presented in cited literature. Within the revised version of the manuscript, the reader is referred to the original paper in literature, for more information.
- Tables and Figures that contain similar data are not necessary. Please leave only Table of Figure.
- We left only Figures
- What is the trend of Figure 4? Please also clarify the definition of normalized cell index.
- The normalized cell index (NCI) was calculated during RTCA experimental measurements, by the software within the xCELLigence RTCA DP system. The NCI is an operation function by which the cell index is established as 1.0 (100%) in the moment of the treatment. After that, all measuring data are expressed in proportion with this measure. This explanation was provided within Section 2.3, but we failed to refer the reader additionally to this Section when writing the Results section. This is now corrected in the revised version of the manuscript, with an additional clarification of this quantity. The trend shown in Figure 4 is also now discussed in the revised version of the manuscript.
Reviewer 3 Report
1. Lines 257-258: what is the meaning of “domain divided into two phases”?
2. Equation (1): Why the same s1 in numerator and denominator? What is the meaning of oxygen dependence in cell death and conditions on the coefficients (increasing or decreasing function)?
3. Equation (2): Quasi-stationary approximation, 1D space derivative – below written 2D
4. Domain and boundary conditions should be explicitly written.
5. Line 287, no boundary conditions needed for the cell concentration.
6. Lines 288-289, what is the meaning of oxygen tension on the boundary and outside cells?
7. Line 325, cells can consume as much oxygen as needed – how this is presented in the model?
8. Lines 354-364: selectivity of different drugs is discussed, that is, how they act on cancer cells in comparison with how they act on healthy cells. The reference is given to Figure S3. However, this figure shows the comparison of treated cancer cells with untreated cells. The figure does not correspond to the text. In general, treatment selectivity towards cancer cell is not clear.
9. Lines 452 and below: how error was estimated? In Figures 2 and 3, on the vertical axis “percentage of change” – what this means, what change? In the text it is written that this is error estimation. The values can reach 2000% - error 20 times? All this is quite confusing. If it is the comparison between different cell lines, this should be clearly explained in the text and in the figure captions.
Author Response
- Lines 257-258: what is the meaning of “domain divided into two phases”?
- The expression was rather imprecise and was corrected in the revised version of the manuscript. The domain is considered to contain two phases
- Equation (1): Why the same s1 in numerator and denominator? What is the meaning of oxygen dependence in cell death and conditions on the coefficients (increasing or decreasing function)?
- The fact that same s1 is contained within numerator and denominator arises from the derivation procedure of the equations of the numerical model, more precisely during the nondimensionalisation. Details of this procedure are provided in the original paper where the numerical model was presented. The reader is refered to the cited literature in the revised version of the manuscript for more details.
- Equation (2): Quasi-stationary approximation, 1D space derivative – below written 2D
- The derivative is 2D space, the x is a vector quantity and should be bolded. In the revised version of the manuscript the error of writing x as a scalar is corrected.
- Domain and boundary conditions should be explicitly written.
- The explicit definition of conditions is included in the revised version of the manuscript.
- Line 287, no boundary conditions needed for the cell concentration.
- Since the simulation domain covers only a portion of the entire experimental domain (not all well plates), we had to define the boundary condition for the cell concentration.
- Lines 288-289, what is the meaning of oxygen tension on the boundary and outside cells?
- This way we ensured that there is enough input of oxygen, i.e. that the cells can consume as much oxygen as needed.
- Line 325, cells can consume as much oxygen as needed – how this is presented in the model?
- This is presented in the model using the boundary conditions, like it is discussed in the previous point of the Reviewer.
- Lines 354-364: selectivity of different drugs is discussed, that is, how they act on cancer cells in comparison with how they act on healthy cells. The reference is given to Figure S3. However, this figure shows the comparison of treated cancer cells with untreated cells. The figure does not correspond to the text. In general, treatment selectivity towards cancer cell is not clear.
- Clarified in the text by better referencing of figures in the text
- Lines 452 and below: how error was estimated? In Figures 2 and 3, on the vertical axis “percentage of change” – what this means, what change? In the text it is written that this is error estimation. The values can reach 2000% - error 20 times? All this is quite confusing. If it is the comparison between different cell lines, this should be clearly explained in the text and in the figure captions.
- It was a rather imprecise explanation in the original manuscript. The estimation errors were calculated using Eq. (3). And the values of these errors for all considered cases varied between 0.03 and 0.5, like it was mentioned in the beginning of Section 3.4. The remaining of Section 3.4 discusses the estimated values of parameters of the model (s1-s4). The values of these parameters vary depending on the treatment. Figures 2 and 3 are plotted with relative values of the parameters. The percentage of change of parameters refers to the relative percentage of the value of parameters for each treatment, with respect to the parameters obtained for the control cell line. In the example the Reviewer mentioned, it means that a parameter for that particular case is 20 times greater for that treatment, in comparison to the value for the control case. This is additionally clarified in the text of the revised version of the manuscript and included in figure captions.
Round 2
Reviewer 1 Report
Accepted
Accepted
Author Response
Thank you very much for kind suggestions. We made changes on some English language mistakes in the text body.
Reviewer 3 Report
The same questions as before:
1. line 267 and below, "oxygen tension" - this is not conventional terminology, do you mean oxygen concentration?
2. line 292, introduction of the boundary conditions for the cell concentration alpha is mathematically incorrect.
3. line 294, "oxygen tension outside cancer cell is constant" - not clear what this means and how it is implemented in equation (2)
Author Response
Dear professor,
Please find enclosed our answers:
- line 267 and below, "oxygen tension" - this is not conventional terminology, do you mean oxygen concentration?
Authors: The reviewer is right, it is better to use the term oxygen concentration. This is corrected in the revised version of the manuscript.
- line 292, introduction of the boundary conditions for the cell concentration alpha is mathematically incorrect.
Authors: The equation is corrected in the revised version of the manuscript.
- line 294, "oxygen tension outside cancer cell is constant" - not clear what this means and how it is implemented in equation (2)
Authors: This condition is ensured by prescribing the oxygen values in the appropriate nodes of the mesh. This is clarified in the revised version of the manuscript.
Round 3
Reviewer 3 Report
I have the same questions as before to the paragraph 284-290. Looks like the authors do not understand what I say.
Author Response
We sincerely apologize, but the authors of this part of the manuscript (TD and Prof. NF) don’t understand what exactly does the Reviewer think is not correct. The boundary conditions were taken from the original model presented in cited literature and were only used accordingly for the present experimental setup.